# Oxygen Scavenger and Antioxidant LDPE/EVOH/PET-Based Films Containing β-Carotene Intended for Fried Peanuts (*Arachis hypogaea L.*) Packaging: Pilot Scale Processing and Validation Studies

**DOI:** 10.3390/polym14173550

**Published:** 2022-08-29

**Authors:** Adriana Juan-Polo, Salvador E. Maestre Pérez, María Monedero Prieto, Carmen Sánchez Reig, Ana María Tone, Nuria Herranz Solana, Ana Beltrán Sanahuja

**Affiliations:** 1Department of Analytical Chemistry, Nutrition and Food Sciences, P.O. Box 99, 03080 Alicante, Spain; 2Packaging, Transport & Logistics Research Center (ITENE), Albert Einstein 1, Valencia, 46980 Paterna, Spain

**Keywords:** oxidative stability, beta-carotene, oxygen absorber film, fried peanuts, antioxidant capacity, active food packaging

## Abstract

The aim of this study was to develop an oxygen scavenger and antioxidant active packaging material for fried peanuts. The packaging solution, which has been made at the laboratory previously, has been developed by cast film extrusion and is composed of low-density polyethylene-ethylene vinyl alcohol-polyethylene terephthalate (LDPE/EVOH/PET)-based films containing β-carotene (CAR). In comparison with film without additive, developed film presented an orange colouring (higher L* and b* values and lower a* values) and an increase in oxygen induction time (OIt) from 4.5 to 14.1 min. The incorporation of β-carotene to the formulation also brings about a significant effect on the thermal stability as maximum degradation temperatures increased around 1%. Regarding the oxygen absorption capacity of the films, values of 1.39 ± 0.10 mL O_2_ per g of film at laboratory scale and 1.7 ± 0.3 mL O_2_ per g of multilayer (ML)/LDPE_CAR were obtained, respectively, after 3 days, proving the suitability of the packaging solutions as oxygen absorbers. To validate the packaging solution, the oxidative stability of fried peanuts packed in fabricated multilayer β-carotene bags was evaluated for 3 months at 40 °C. The hexanal content remained constant during this period. Meanwhile, peanuts packed in ML without β-carotene increased their hexanal content to 294%. This fact indicated a lower extent of oxidation in fried peanuts compared to food samples packaged in control films, suggesting the potential of ML/LDPE_CAR films as sustainable and antioxidant food packaging systems to offer protection against lipid oxidation in foods. Sensory evaluation confirmed that ML/LDPE_CAR films provided the peanut samples with an extra aroma due to the volatile degradation products of β-carotene (such as β-cyclocitral or 6-methyl-5-hepten-2-ol).

## 1. Introduction

Food spoilage processes that occur after harvesting and during processing and transport are unavoidable. Even after processing to reduce the microbiological burden, various chemical and biological processes continue in foods and determine their shelf life. To reduce food waste, the food industry is using tools such as improving processing practices and developing novel packaging solutions, among others.

Fat oxidation is one of the most important degradation reactions that affect food quality, food safety and consumer’s perception as the oxidation process degrades fatty acids generating toxic aldehydes and decreasing the quality of the food product [1]. In this sense, the shelf life of food products is of a major concern for the food industry. This is especially important for the nuts industry, as most of the nuts contain high levels of fat.

Peanuts (*Arachis hypogaea*) have roughly 50 wt.% of fat, with unsaturated fatty acids such as oleic and linoleic fatty acids accounting for more than 80% of the fat fraction. Since the susceptibility of unsaturated fatty acids to oxidation reactions is higher than that of saturated fatty acids, this composition makes peanuts particularly sensitive to oxidation. Other variables that affect lipid oxidation are temperature, water content, free fatty acid content, oxygen concentration, and the presence of prooxidant and antioxidant substances [2]. In this line, tocopherols present in peanuts (mainly α- and γ-tocopherol homologues) act as an antioxidant barrier and protect unsaturated fatty acids from suffering oxidative damage [3]. When peanuts are being oxidised, the concentration of some volatile compounds such as hexanal, a rancidity indicator, increase as it is a secondary metabolite of lipid oxidation [4,5]. Consequently, hexanal analysis is a useful tool to monitor oxidative damage in peanuts [6]. Nowadays, when peanuts are packed, the control of the oxygen concentration inside the packaging is achieved through the use of polyethylene/polypropylene (PE/PP). PE/PET multilayer films, among others, containing EVOH as an oxygen barrier layer [7]. Moreover, the modified atmosphere packaging (MAP) technology is also used but, even with these packaging solutions, there is an amount of residual oxygen (around 0.5–1 vol.%) inside the container that cannot be avoided. This accelerates oxidation reactions, reducing the shelf-life of peanuts [8]. For all these reasons, the use of active packaging has become a promising approach to the conventional packaging solutions. One type of active packaging consists of the direct addition of oxygen scavengers into packaging materials, therefore reducing further the concentration of oxygen in the packaging atmosphere (i.e., less than 1 vol.%) and extending the shelf life of the packed products. 

Currently, there are several types of available oxygen scavengers. According to Gaikwad et al. [9], oxygen scavengers can be classified in: metallic, inorganic, organic, polymer, and enzyme-based compounds. Metallic-based scavengers are compounds based on iron, palladium, or other transition metals that require a certain water activity in the food to provide moisture to facilitate the oxygen removal by the scavenger [10]. Since they need the presence of water, they are not suited for their use when low moisture foods are packaged. They are by far the most frequently used group, specifically those relying on iron salts. In recent years, much attention has been devoted to the development of oxygen scavengers of the other groups. In the organic-based group, compounds such as butylated hydroxytoluene (BHT), α-tocopherol, or different flavonoids (catechin, quercetin…) have been evaluated [4,11,12,13]. 

In the recent literature, some oxygen scavengers have been developed for their use in packaging for peanuts. They include the incorporation of trans-polyisoprene, with cobalt chloride as a metallic catalyst, and polybutadiene to polyethylene films [14,15]. However, the two systems need to be activated previously by using UV radiation, hindering their implementation in processing packaging plants since an additional step is needed.

Consumer’s preferences have steadily evolved to have a preference for natural over synthetic products because natural is seen as safer. Examples of naturally derived oxygen scavenging compounds are ascorbic acid and its salts, isoascorbic acid, tocopherol, hydroquinone, catechol, sorbose, lignin, gallic acid, and polyunsaturated fatty acids [16]. Unsaturated fatty acids can be used as oxygen absorbers without needing the presence of moisture for their activation, making them useful for low moisture food applications. However, the absorption mechanism involves a catalytic autoxidation reaction that generates aldehydes producing unpleasant odours. 

For peanut packaging, the use of sodium ascorbate in PE films was investigated by Modaresi and Niazmand [17]. The developed film was able to reduce the lipid oxidation process in peanuts using a GRAS (“Generally Recognized As Safe”) categorised compound. Nevertheless, due to the characteristics of sodium ascorbate, the mechanical properties and water permeation of films were modified. Flavonoids have also been incorporated into different polymeric matrices [4,6,18]. It is important to note that film functionality is frequently verified by using traditional antioxidant capacity methods such as 2,2-diphenyl-1-picrylhydrazyl (DPPH) or 2,2′-azino-bis(3-ethylbenzothiazoline-6-sulfonic acid) (ABTS) methodologies, but actual oxygen absorption is not verified.

β-Carotene is a natural pigment characterised by its conjugated double bond system. It is well-known as a UV stabiliser and as a singlet oxygen quencher [19]. These properties can be useful for food preservation in the food industry. The antioxidant and preservative characteristics of carotenoids have been thoroughly studied in food products. In the packaging sector, β-carotene has received far less attention, probably because of its pigmentation capacity, although its use as a polymer stabiliser has been evaluated. Hence, López-Rubio et al. found that 2 × 10^−4^ g β-carotene/ mL of polymer film increases their UV stability in various biopolymers including poly(lactic-acid) (PLA), poly(ε-caprolactone) (PCL), and poly(hydroxybutyrate-co-valerate) (PHBV) [20]. The additive also plasticized the biopolymer. This fact has also been demonstrated in a PE matrix. Abdel-Razik confirmed that the addition of β-carotene into an acrylonitrile-butadiene-styrene (ABS) matrix stabilised the copolymer due to its capability as a singlet oxygen quencher [19]. Moreover, it was also demonstrated β-carotene hindered oxidation process in the propagating step of the thermo-oxidation of ABS (interfering in the formation of peroxy and alkoxy radicals) because β-carotene levels decreased at the beginning of the thermal treatment.

From the point of view of the effect of materials containing β-carotene on packaged foods, this compound has been used in materials developed to contain oils such as soybean and sunflower oils [21,22]. Nevertheless, to our knowledge, it has not been employed for solid foods yet. Thus, the aim of the present study is the development of a double-function oxygen absorber and antioxidant films for active packaging by the incorporation of β-carotene into LDPE/PET based polymer matrices, at laboratory and pilot-scale. The effects of the addition of β-carotene on the oxygen absorption capacity, optical, and thermal properties of the films were studied. Moreover, the antioxidant capacity of the active films was evaluated by the oxidation induction time (OIt) measurements by differential scanning calorimetry (DSC). Overall migration tests were also analysed. In order to validate the developed materials, the efficiency of the films was evaluated by conducting a shelf-life-study with fried peanuts stored for 3 months at 40 °C.

## 2. Materials and Methods

### 2.1. Materials and Reagents

Peanuts (20 kg) were purchased in a Spanish food store, and they were stored in packaging (1 kg batches) in nitrogen atmosphere at 25 °C and controlled light exposure. A portion of the sample (2 kg) was preserved at −20 °C until sample analysis to have a fresh sample.

LDPE ALCUDIA^®^ 2221FG was purchased from Repsol (Madrid, Spain). A commercial multilayer film with a structure relation of PET/EVOH/LDPE Multilayer material (coded by ML) was obtained from VIDUCA S.L. (Alicante, Spain).

β-carotene was supplied by Nanjing NutriHerb BioTech Co., Ltd. (Nanjing, China). N-hexane (99% GC grade) and isopropanol (HPLC) grade were purchased from Panreac (Barcelona, Spain). Petroleum ether was acquired from Sigma–Aldrich Inc. (St. Louis, MO, USA).

### 2.2. Films Processing 

LDPE-based films with β-carotene were prepared by cast film extrusion. Two formulas of packaging materials were obtained. Firstly, monolayer packaging (LDPE_CAR) was processed at laboratory scale. Secondly, a multilayer packaging (ML/LDPE_CAR) was produced at pilot plant scale. 

LDPE_CAR: LDPE and β-carotene (10 wt.%.) were melt-mixed using a Brabender DE 20/40D co-rotating twin-screw extruder (Plastograph, Dusseldorf, Germany). Mixing was done in a batch internal mixer Brabender at 40 rpm at five temperature stages (temperatures of 185, 180, 175, 170, and 170 °C, respectively). Control sample was LDPE without any additive.

ML/LDPE_CAR: The structure of the multilayer film was processed into two steps. In the first step, β-carotene (10 wt.%) was incorporated into the LDPE matrix by a co-rotating twin screw extruder Leistritz ZSE 27 MAXX (Leistritz Pumps GmbH, Nuremberg, Germany) with a length-diameter ratio (L/D) of 44. To process the β-carotene compound, the machine had a gravimetric feeder. The temperature profile set along the 12-barrel zones was 60/80/180/180/180/180/180/180/180/180/180/180 (°C) with an extrusion speed of 250 rpm and a throughput of 17 kg/h. In a second step, a commercial multilayer film (Viduca, Spain) with a structure relation of PET/EVOH/LDPE was laminated with a co-extrusion coating of a layer film consisting of LDPE β-carotene with a co-extrusion line, Dr. Collin MF-EXB-600 (COLLIN Lab & Pilot Solutions, Ebersberg, Germany) based on three single co-extruders (E30P 25L/D). For the layer film processing, the extruders were fed with the masterbatch obtained in the first step. The temperature profile set for extruder B was 170/205/215/220/225 (°C) and for extruder C, it was 190/210/215/220/220 (°C). 

The developed film samples were stored in hermetically sealed aluminium/LDPE bags and maintained at −20 °C until the analysis. An ID-S112B Digimatic Micrometer (Mitutoyo, Japan) was employed to measure the thickness of every sample at 25 °C. Measurements were made at five arbitrary positions, obtaining average thickness values of 98 ± 5, 133 ± 6; 74 ± 1, and 122 ± 2 µm for LDPE, LDPE_CAR, ML, and ML/LDPE_CAR, respectively.

### 2.3. Films Characterization

#### 2.3.1. β-Carotene Quantification

The ML/LDPE_CAR films (0.3 g) were stirred in 20 mL of hexane at 60 °C for 24 h in the dark to extract β-carotene. BHT was added to the solution (0.4 mg mL^−1^) in order to avoid β-carotene degradation during the extraction period. The extracts were analysed by UV-Vis spectroscopy (Biomate 3S, Thermo Fisher Scientific, MA, USA) at 466 nm. Calibration curve was prepared by dissolving different amounts of β-carotene in hexane with BHT. Β-carotene content was expressed in mg β-carotene per g of film. Three replicates were performed.

#### 2.3.2. Instrumental Colour Analysis

The colour of all packaging materials was measured by using a spectrophotometer Konica CM-3600d (Konica Minolta Sensing Europe, Valencia, Spain) by using the CIELAB colour notation system (International Commission on Illumination, 1976, Vienna, Austria). The *L**, *a**, and *b** represent three dimensions of a measured colour, which gives specific colour values of the material. 

The total colour difference (Δ*E*) with respect to the reference film was calculated from three colour coordinates following Equation (1). All samples were analysed in triplicate.
(1)$$\Delta E=\sqrt{{\left(L_{REF}^{*}-L^{*}\right)}^{2}+{\left(a_{REF}^{*}-a^{*}\right)}^{2}+{\left(b_{REF}^{*}-b^{*}\right)}^{2}}$$
where *L*_REF_, a*_REF_*, and *b*_REF_* values are parameters from control films (LDPE or ML) and *L*, a*,* and *b** are parameters from LDPE/CAR or ML/LDPE_CAR.

#### 2.3.3. Thermal Analysis

DSC analysis was carried out by employing a TA Instruments Q100 equipment (TA Instruments, New Castle, DE, USA) working in nitrogen atmosphere (50 mL min^−1^). Film samples (5.0 mg) were added in aluminium pans (70 µL) and the applied thermal programme was: Heating from 25 °C to 300 °C (3 min hold), cooling to 0 °C (5 min hold), and heating to 300 °C; all steps at 10 °C min^−1^. DSC were obtained from the first heating scan. Crystallisation and melting temperatures (T_c_ and T_m_) were calculated at peak temperatures of the corresponding transitions, while the crystallisation and fusion enthalpies (∆H_c_ and ∆H_m_) were obtained from the area of the corresponding peaks. The crystallinity index (*X*_c_) of LDPE was obtained by using the Equation (2):*X*_c_ = (ΔH_m_/ΔH°_m_W_LDPE_) × 100(2)
where ΔH_m_ is the melting enthalpy per unit mass of the sample, ΔH^°^_m_ is the theoretical value of the melting enthalpy per unit mass of 100% crystalline PE (293 J/g) [23], and W_LDPE_ is the weight fraction of the LDPE matrix.

The OIt was determined by using the same DSC equipment described above. Five mg of multilayer films (ML and ML/LDPE_CAR) were enclosed in a 70 µL aluminium pans and they were heated to 200 °C at 10 °C min^−1^ in nitrogen atmosphere (50 mL min^−1^). After that, samples were maintained at 200 °C during 30 min under the air atmosphere. The beginning of material oxidation was clear due to an increase of the slope of the exothermal heat flow. The time before switching to air was removed and OIt was the time in which the slope starts to increase [24].

Thermogravimetric analysis (TGA) experimental determinations were performed by employing a TGA/SDTA851e/SF/1100 Mettler Toledo (Mettler Toledo, Schwarzenbach, Switzerland) equipment. Films (5.0 mg) were heated from 30 to 700 °C at 10 °C min^−1^ in nitrogen atmosphere (50 mL min^−1^). The initial degradation temperature (T_ini_), calculated at 5% of weight loss, and the temperature of maximum decomposition rate (T_max_) were obtained. Triplicate DSC and TGA analysis of all the samples were carried out.

#### 2.3.4. Oxygen Absorption Capacity 

Initially, oxygen absorption of LDPE_CAR and ML/LDPE_CAR films was tested during a week to fix the suitable day to measure absorption capacity. Therefore, 0.25 g of active films were placed into a 20-mL sealed vial and kept at 30 °C in an incubator (Inkubator 100, Heidolph Instruments GmbH & CO, Schwabach, Germany). The oxygen contents inside the vials were measured after 1, 2, 3, 4, and 7 days of exposure by a headspace O_2_/CO_2_ analyser OXYBABY® 6.0 (WITT-Gasetechnik GmbH & Co KG, Witten, Germany) by puncturing the probe through a foam rubber seals. 

Then, different amounts (two replicates of 0.25, 0.50, 0.75, and 1.00 g) of both references were incubated at the optimum exposition time, and then they were subjected to the mentioned procedure. Oxygen absorption values were expressed as mL oxygen g^−1^ film.

#### 2.3.5. Overall Migration Tests

The overall migration tests were, carried out in ML/LDPE_CAR films. To carry out the tests, simulants A (ethanol 10 % (*v*/*v*) and D2 (vegetable oil) were selected according to Annex III of the Plastics Regulation [25]. However, in this work isooctane and ethanol 95% (*v*/*v*) were employed as alternatives to simulant D2. Contact conditions OM2 (10 days at 40 °C) were selected according to Annex V of the Plastics Regulation and considering the use of the material. 

The migration test was carried out by using single-side contact of 50 cm^2^ circular strips with 25 mL of the food simulant. Samples in ethanol (10% and 95% (*v*/*v*)) were maintained in an oven (Selecta, Barcelona, Spain) at 40 °C for 10 days. On the other hand, samples contained in isooctane were kept at 20 °C for 2 days, according to the EN-1186-5 standard [26]. After the contact time, films were taken out and the simulants were entirely evaporated by obtaining the residual compounds, which were determined gravimetrically. The overall migration value was obtained in mg dm^-^^2^, referring to the migration of the compounds (mg) in a standard contact area of 1 dm^2^. Triplicate overall migration analysis of all the samples was carried out.

#### 2.3.6. Volatile Fraction of Films

The volatile fraction of films was analysed by headspace-solid-phase microextraction (HS-SPME) coupled to gas chromatography–mass spectrometry (GC–MS) using an Agilent 7000 triple quadrupole mass analyser (QMA) (Agilent, Palo Alto, CA, USA). An amount of 1 g of each sample was introduced in a 20 mL vial and sealed with an aluminium crimp cap with a polytetrafluroethylene/silicone septum. The vial containing the sample was moved by using a mechanical arm present in the autosampler to an oven to be heated at 85 °C for 60 min. After that, a SPME fibre (Divinylbenzene/Carboxen/Polydimethylsiloxane (DVB/CAR/PDMS) SPME, 50/30 m, StableFlex, 1 cm long) located on the mechanical arm was exposed 20 min to the sample HS to extract the volatile substances. Afterwards, the fibre was withdrawn and introduced into the GC injector for desorption (5 min) and analysis. GC-MS injection port was set at 250 °C in splitless mode. A DB-624 column (30 m × 320 μm × 1.8 μm; Agilent, CA, USA) was used, which was programmed from 40 °C to 250 °C (hold 1 min) at 10 °C min^−1^. Helium was used as carrier gas (2 mL min^−1^) and analysis in triplicate were carried out for all the samples. 

### 2.4. Packaging of Fried Peanuts

Multilayer materials (with and without β-carotene) were employed to make 25.5 × 9 cm bags (Figure 1). A piece of film sample (26 cm × 20 cm) was folded and heat sealed, remaining an open side. Fresh fried and peeled peanut samples (250 g) were loaded into the pouch, and the opening of the pouch was heat sealed also. Atmosphere into the bags was air, so final oxygen amount was around 70 mL per bag.

Bags were stored at 40 °C for 3 months. Each month, 3 bags of each reference were opened and mixed. Peanuts inside the bags were coded as P_ML and P_ML/LDPE_CAR. Two kg of peanuts were stored at −21 °C as a control sample (P_Control) in their original package. The hexanal content and sensory analysis were performed every month.

### 2.5. Oxidative Stability Study of Packaged Fried Peanuts

#### 2.5.1. Hexanal Content

Hexanal is a volatile formed as a consequence of the peroxidation reaction of linoleic acid in nuts [6]. In order to monitor the oxidative degradation of the peanuts packed in the developed films, the analysis of hexanal inside the packaging was carried out [4,5]. The hexanal was analysed by HS-SPME-GC-MS by using an Agilent 6890N GC System (Agilent, Palo Alto, CA, USA) coupled to an Agilent 5973N triple quadrupole mass spectrometry (Agilent, Palo Alto, CA, USA) equipment. A TDS-2 Gerstel MultiPurpose autosampler (, Gerstel GmbH, Mülheim an der Ruhr, Germany) was also used.

Samples (0.5 g of ground peanuts) were heated at 70 °C for 30 min. After the heating process, the SPME fibre (DVB/CAR/PDMS SPME, 50/30 m, StableFlex, 1 cm long) located on the mechanical arm was exposed to the sample vial HS to extract the volatiles at 70 °C, for 15 min. Then, fibre was desorbed for 3 min at 250 °C into the GC injector port by using the splitless mode. A DB-624 column (30 m × 250 μm × 1.4 μm; Agilent, CA, USA) was employed and it was arranged from 50 °C to 250 °C (hold 12 min) at 10 °C min^−1^. Helium was employed as a carrier gas (1 mL min^−1^). MS data were recorded between 30–550 m/z with an electron energy of 70 eV. The temperature values of the ion source and the transfer line were 230 and 150 °C, respectively. National Institute of Standards and Technology (NIST) was employed as identification library and triplicate analysis were performed for all the samples.

#### 2.5.2. Sensorial Analysis 

Sensory analysis was performed using a 40-member panel, 22 males and 18 females, with ages between 21 and 64 years. Panellists were not trained, and verified they were available for all sensory evaluation sessions. During the sensory evaluation, water was provided to rinse mouths between samples.

A sensory evaluation was carried out each month (during 3-month assay). Firstly, two triangular tests were performed to evaluate overall perception among three samples (P_control, P_ML, and P_ML/LDPE_CAR). First triangular test confronted P_control and P_ML samples and second triangular test confronted both thermal treated samples (P_ML and P_ML/LDPE_CAR). The test consists of tasting two equal samples and one different sample and identifying the different one. Six peanuts per treatment were served at room temperature in plastic bags labelled with 3-digit random codes. 

In a second stage, panellists had to rank three peanut samples (P_control, P_ML and P_ML/LDPE_CAR) in order of hardness and colour intensity. Hardness was previously defined as the force required to compress a substance with the teeth. Results were evaluated by Friedman test (UNE-ISO 8587). In Equation (3), *R* is the sum of ranks of the product I and *p* and *j* are the number of samples and the number of judges, respectively.
(3)$$F_{test}=\frac{12}{j\cdot p\left(p+1\right)}\left(R_{1}^{2}+\ldots +R_{p}^{2}\right)-3j\left(p+1\right)$$

Differences between three samples are significant if *F_test_* is higher than *F* (*j* = 40, α = 0.05).

### 2.6. Statistical Analysis

Statistical analysis of experimental data was performed with R software (version 4.2.0). The data is represented as means ± standard deviation. Differences between replicates of mean values were evaluated by the Tukey test for α = 0.05 (confidence interval of 95%).

## 3. Results and Discussion

### 3.1. Characterization of Films Processed at Laboratory Scale

LDPE and LDPE_CAR monolayer films were processed at laboratory scale to evaluate the viability of β-carotene incorporation in a LDPE matrix. Firstly, colour characteristics of the films were determined. 

Table 1 shows that *a** and *b** values were higher in LDPE_CAR than in LDPE confirming that the active compound incorporation causes, as it is described elsewhere [21,27], an intense yellow-orange colour. In addition, a decrease of around 25% in the value of *L** parameter was observed when β-carotene was present in the LDPE_CAR formulation, indicating an increase in the opacity of the films in comparison with the control one. These results are in line with those obtained in other studies carried out by Beltrán et al. [28], in which a significant decrease in *L** value was identified as a consequence of the incorporation of hydoxytyrosol as antioxidant compound in polycaprolactone films. 

In relation to the thermal properties of the films, the shape of the DSC melting curve depends on the thermal history of the sample, thus a peak is observed at 108 °C related to the melting of LDPE in LDPE and LDPE_CAR samples, as it was reported in previous studies [19,29]. In relation to the crystallization process, values of T_c_ around 97 °C are obtained in both materials and significant differences were observed in the melting and crystallization enthalpies. Regarding the TGA results, the initial degradation temperatures (T_ini_), determined at 5% weight loss, and maximum degradation temperatures (T_max_) obtained for all formulations are shown in Table 2. The incorporation of β-carotene to the formulation brings about a significant effect on the thermal stability of the obtained active films (*p* < 0.05). In this sense, the addition of a theoretical amount of 10 wt.% β-carotene to LDPE films resulted in an increase in the T_max_ of 3%. 

Finally, oxygen absorption capacity of LDPE_CAR film was evaluated. As can be seen in Figure 2, LDPE_CAR film was able to absorb 1.39 ± 0.10 mL O_2_ per g of film. An increase in the absorbed oxygen was observed as the amount of film increase from 0.5 to 1.5 g confirming a linear behaviour when the absorbed oxygen (mL) was represented versus the film mass (g). No previous studies have been found related to the incorporation of β-carotene to LDPE films with the purpose to act as an oxygen scavenger, without the need of any additional activation process. All the obtained results confirmed that β-carotene incorporation in LDPE matrices at laboratory scale produces valuable films in terms of thermal properties and oxygen absorption capacity.

### 3.2. Characterization of Films Processed at Pilot Scale

#### 3.2.1. Quantification of β-Carotene

β-Carotene content was quantified in ML/LDPE_CAR films and the result was 0.63 ± 0.09 g β-carotene per 100 g film. Considering that the thickness of the LDPE with β-carotene layer was 25 µm and the ML/LDPE_CAR had a total thickness of 122 µm, the incorporated concentration of β-carotene after the processing was 3.07 wt.%. 

Similar loss percentages (around 70%) were obtained in the β-carotene extrusion process [21]. These losses are usual when natural antioxidants are exposed to high temperatures. For example, C. Wessling et al. reported losses of 66% when α-tocopherol was extruded in LDPE matrix [30]. 

#### 3.2.2. Colorimetric Analysis

To evaluate the effect of the presence of β-carotene in the pilot-scale developed films, colorimetric analysis was carried out, since colour changes are also employed in developing functional packaging materials with indicator function. Significant differences in colour were observed as the result of the β-carotene addition in ML samples (*p* < 0.05), as can be observed in Table 1.

In this sense, neat ML showed high *L** value, with a significant decrease because of the presence of the additive. This indicated a significant darkening of these films, as was also observed for the LDPE_CAR samples. Some authors have also reported a similar effect when adding antioxidant additives to polymer matrices [18,20,31]. Significant differences in *a** and *b** values were also found in films containing β-carotene in comparison with control samples (*p* < 0.05) resulting in a slightly amber colour. ML/LDPE_CAR presented higher *b** value than ML. However, in contrast to LDPE_CAR films, *a** value was lower. It could be explained because of different thickness between two films with β-carotene [20].

Similar colour changes were reported for polypropylene stabilised with hydroxytyrosol as antioxidant, contributing to strengthening the colour of the obtained films [28,32].

#### 3.2.3. Thermal Properties

The thermal properties of the pilot scale processed films were evaluated in terms of DSC and TGA analysis. DSC parameters for multilayer LDPE based films containing β-carotene and for the control films are shown in Table 3. In ML and ML/LDPE_CAR samples, two endothermic peaks were obtained at about 102–105 °C and 111 °C, respectively, related to the melting process of LDPE, which can be attributed to several factors such as two different crystallographic forms, crystallites of varying degrees of perfection, and differences in crystallite size, among others [19,33]. The same behaviour was observed in the crystallisation process, and significant differences were obtained in this parameter when comparing the control films with the ones containing β-carotene. In these multilayer samples, two additional melting peaks were distinguished at around 180 °C and 254 °C, respectively, attributed to the presence of EVOH and PET, as was expected [19,23]. 

In relation to the role of β-carotene as a protector against polymer oxidation, in previous works it has been postulated that the stabilising effect of it depends upon their ability to form stable products during the thermooxidative degradation of different polymers such as ABS, among others [19]. This type of stabiliser is considered to have a multifunctional role in many polymers. Especially for polyolefins, OIt tests are well established for quality control purposes as a quick screening method to check the performance of the stabilisation systems. This induction time can be related to the amount of antioxidant present in the polymer or the effectiveness of any particular antioxidant in the formulation [32]. The addition of β-carotene to the studied multilayer LDPE based films resulted in a significant 9.6 min increase in the OIt when compared to the material without the antioxidant. This effect could be attributed to the amount of β-carotene that remained in the films after the extrusion process (3.07 wt.%). This is a clear indication of the good stabilisation obtained with the addition of low amounts of β-carotene to polyolefins, as indicated by other authors who studied the effect of natural antioxidants as polyolefins stabilisers [24]. 

The thermogravimetric results confirmed a principal degradation step for all the analysed samples. The TGA T_ini_ and T_max_ values related to all the analysed samples are presented in Table 3. The incorporation of β-carotene to the formulation brings about a significant effect on the thermal stability of the obtained active films (*p* < 0.05). In this sense, the presence of 3 wt.% β-carotene to ML_LDPE films resulted in an increase in the T_max_ of 1%. 

#### 3.2.4. Oxygen Absorption and Kinetics

The oxygen scavenging capacity of the ML/LDPE_CAR films developed by cast film extrusion at pilot scale was also evaluated, since oxygen content accelerates food spoilage in fatty food. In a preliminary test, the O_2_ absorbed per film amount was monitored for a week, performing a measurement each day. Figure 3 shows that β-carotene films absorbed the oxygen present in the headspace from the first day to the third day of exposure to air. From then on, their oxygen absorption capacity remained constant. For this reason, the oxygen measurements from the headspace of the 20 mL-vials were carried out after three days of the air exposure. 

The amount of oxygen absorbed (mL) by ML/LDPE_CAR films after three days at 30 °C was evaluated. As it is shown in Figure 2, oxygen absorption capacity was 1.7 ± 0.3 mL O_2_ per g of ML/LDPE_CAR. It is important to notice oxygen absorption capacity between LDPE_CAR and ML/LDPE_CAR film did not show significant differences, in spite of thicknesses being different. Reactions between oxygen and β-carotene inside the films depend on the concentration of β-carotene and the thickness of the film. D. Tátraaljai et al. verified the β-carotene decomposition time in light was significantly longer in 1 mm thickness films than in 100 µm thickness films (independently of the initial concentration) [34]. To the best of our knowledge, the oxygen absorption capacity of natural compounds such as β-carotene incorporated in a polymeric matrix had not been tested before. ML/LDPE_CAR films were employed to pack 200 g of fried peanuts. When food is packaged under a modified atmosphere or in vacuum, residual oxygen levels from 0.3 to 3.0% remain inside the package [35]. Then, as the developed bags absorb 3.7% of oxygen, β-carotene concentration at 10 wt.%. was appropriated for this application. However, peanuts were packaged with air to accelerate the oxidation process.

#### 3.2.5. Overall Migrations 

The results of the overall migration tests carried out are shown in Table 4 for the three food simulants used and for all the developed materials. A row with the values of the ML material without the layer containing β-carotene has also been included for the sake of comparison. Obtained results indicated that in all tested conditions, the determined overall migration values are below the established limit of 10 mg dm^−2^. It is important to note that migration values obtained using the alternative simulants were corrected according to the legislation (REGLAMENTO (UE) No 10/2011).

A Tukey test for multiple pairwise comparisons (confidence level 95%) was applied to evaluate differences in overall migration data. Results indicated that, related to the ML films, the obtained overall migration values by using isooctane as simulant were higher than those registered with the ethanol-based simulants. When the material containing the layer with β-carotene was tested, results for the overall migration by using 10 % (*v*/*v*) ethanol was lower than those attained with both 95% (*v*/*v*) ethanol and isooctane. As it was expected, higher overall migration values were obtained for the films containing β-carotene in comparison with the control ones. It should be stressed that during the migration assays, the layer that was in contact with the simulant was that containing β-carotene, hence the increase of the overall migration values could be attributed to material migrating from this layer.

The obtained results are in line with previous works which describe the addition of an Azteca marigold extract, rich in carotenoids, in mono and bilayer polyethylene films, and its diffusion into 95% (*v*/*v*) ethanol simulant [36]. The described films had a positive effect on soybean oil oxidative stability as a result of the migration of the carotenoids, among them astaxanthin, present in the films [21].

In this line, a positive effect of the developed films on the oxidative stability of fatty foods was expected because of the higher amount of carotenoids migration in ethanol 95 % (*v*/*v*) and isooctane in comparison with ethanol 10% (*v*/*v*), as it can be observed in Table 4. The obtained results highlight the importance of food simulant selection in the analysis of food-packaging interaction depending on the final application of the developed materials. 

### 3.3. Oxidative Stability of Fried Peanuts Packed in the Developed Films

#### 3.3.1. Hexanal Content in Peanut Samples

The oxidative stability of packaged peanuts (P_ML and P_ML/LDPE_CAR) was determined by the measurement of hexanal content. This compound is related to the oxidation of linoleic acid and used as a rancidity indicator [4]. Figure 4 shows that the hexanal content increased during storage in P_ML peanuts. The higher hexanal relative area values were reached after 2 months of storage in ML bags. In P_ML/LDPE_CAR, the hexanal level was constant over the three evaluated months. 

As the oxidation process in peanuts took place in a larger extension when they were packaged into ML bags compared to ML/LDPE_CAR bags, the oxidation of ML/LDPE_CAR bags was analysed. β-carotene degradation leads to a bleaching of the colour and, thus, colour changes in films can also be used as a method to follow oxidation reactions. Then, the colour of the ML/LDPE_CAR bag was measured after 3 months of thermal exposition to evaluate film oxidation. The colour parameter ΔE did not significantly change in comparison with the initial one, obtaining a value of 13.6 ± 1.3. Therefore, as carotene films loss their strong colour when they are oxidised [34], we can expect that the shelf-life of developed films is longer than 3 months.

In a similar work, Jensen et al. [37] packaged walnuts in LDPE films with atmospheric air at 21 °C. Hexanal content did not increase significantly until 5 months. Hexanal content in walnuts, which were in packages with an oxygen absorber, remained without major changes.

A similar behaviour was also reported by Valdés et al. [38] for the neat poly(Ɛ-caprolactone) (PCL) films and materials containing almond skin extract as antioxidant, in which neat PCL showed higher hexanal content at the end of the study, which can be related to a higher rancidity of packaged food. Based on the obtained results, the addition of β-carotene into the packaging material promotes a lower extent of oxidation in fried peanuts compared to food samples packaged in control films, suggesting the potential of ML/LDPE_CAR films as sustainable and antioxidant food packaging systems to offer protection against lipid oxidation in foods.

#### 3.3.2. Sensorial Analysis

Peanuts were evaluated by a panel composed by 40 panellists. At first, two samples of P_control peanuts (A) and one of P_ML peanuts (B) (coded randomly) were presented to the panellists each month. The obtained results show that there are significant differences between both peanut samples at all the studied times (Figure 5A–C). Considering P_ML presented an increment of 175% in hexanal content after 1 month of thermal treatment (Figure 4), we could conclude that this increment is enough to produce considerable sensory changes.

Then, it is possible to confirm that the increase in hexanal area after 1 month at 40 °C of thermal exposure is sensory detected. Walnuts packaged in LDPE films tasted “very rancid” when hexanal content increased after months in atmospheric air [37].

The second triangle test compared both packaged samples (P_ML and P_ML/LDPE_CAR) during the three studied months (Figure 5D–F.). In every case, the majority of panellists identified which the different sample was. The general comment was that peanuts in P_ML/LDPE_CAR samples tasted distinct, like they were spiced. In order to evaluate the compounds that bring flavour to the sample, the analysis of volatile compounds from the films was carried out. Figure 6 shows the chromatograms of ML and ML/LDPE_CAR films related to the volatile profile. ML film contained mainly the plasticizer isosorbide, which is derived from cellulose or starch [39]. The principal volatile compounds present in ML/LDPE_CAR films were β-cyclocitral (12.8%), 6-methyl-5-hepten-2-ol (11.2%), and dihydroactinidiolide (11.1%). All of them are studied degradation products of β-carotene. β-ionone and β-cyclocitral were generated from the cleavage of (9′,10′) and (7′,8′) bonds of β-carotene, respectively [40]. Dihydroactinidiolide is a secondary oxidation product of the β-ionone [41]. Both compounds are widely used as fragrances due to the sweet odours they produce [41,42].

Finally, there were no significant differences between the three tested samples, either in hardness or in colour attributes (data not shown). Then, β-carotene, as a pigment, did not modify the colour of the samples and the thermal procedure did not dye the samples either.

## 4. Conclusions

Oxygen scavenger and antioxidant films composed by LDPE/PET and β-carotene were successfully formulated and scaled-up, including industrial processing extrusion techniques to obtain commercial products. The incorporation of β-carotene had a positive effect on the oxygen scavenger capacity and thermal stability of the resulting materials, proving the suitability of this innovative packaging solution to extend the shelf-life of fat foods.

In relation to hexanal analysis, and the organoleptic evaluation of packaged fried peanuts, it was observed that active films effectively delayed their oxidative degradation. In conclusion, antioxidant films based on LDPE/PET, containing β-carotene at around 3 wt.%, could be presented as suitable active packaging systems for food preservation, also being an interesting approach to incorporate natural active compounds such as β-carotene that can be obtained from agro-industrial subproducts and wastes, contributing to the circular economy concept.

## Figures and Tables

**Figure 1 polymers-14-03550-f001:**
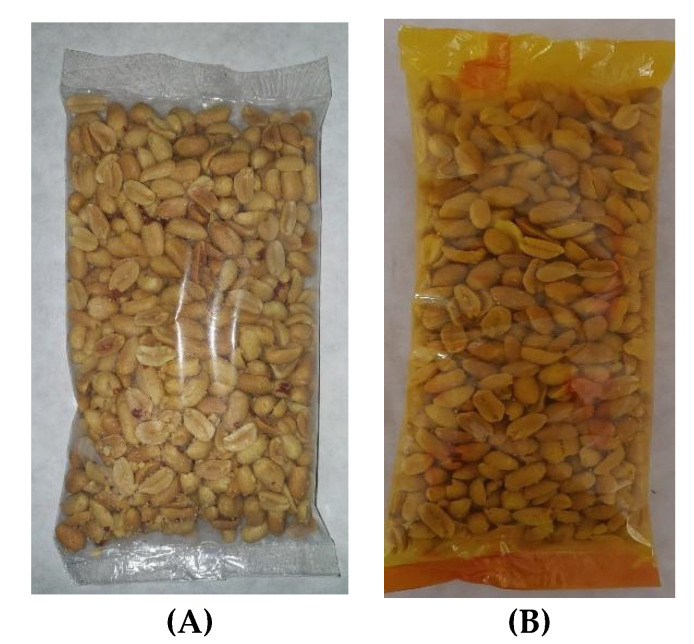
References: (**A**) ML; (**B**) ML/LDPE_CAR.

**Figure 2 polymers-14-03550-f002:**
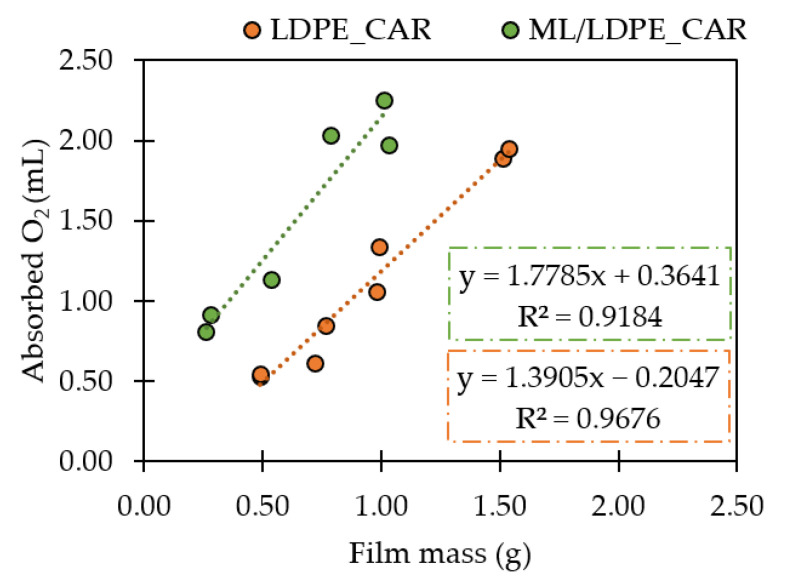
Oxygen absorption of LDPE_CAR and ML_LDPE_CAR films.

**Figure 3 polymers-14-03550-f003:**
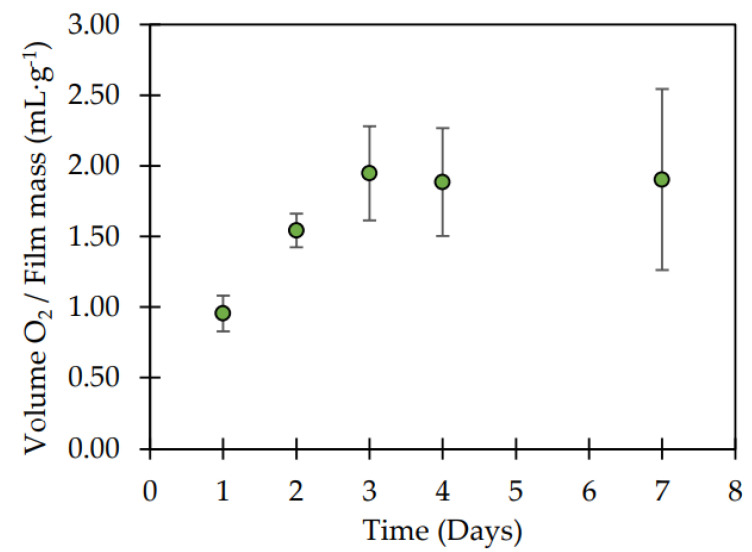
Oxygen absorption kinetics of ML/LDPE_CAR film.

**Figure 4 polymers-14-03550-f004:**
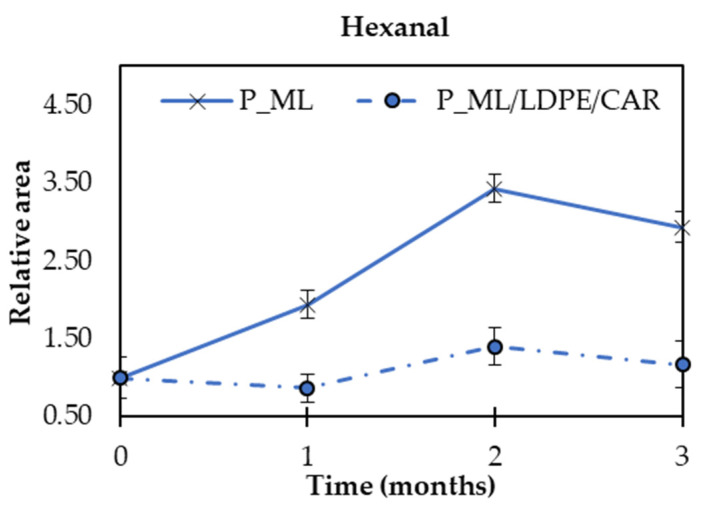
Hexanal content in P_ML and P_ML/LDPE_CAR peanut samples.

**Figure 5 polymers-14-03550-f005:**
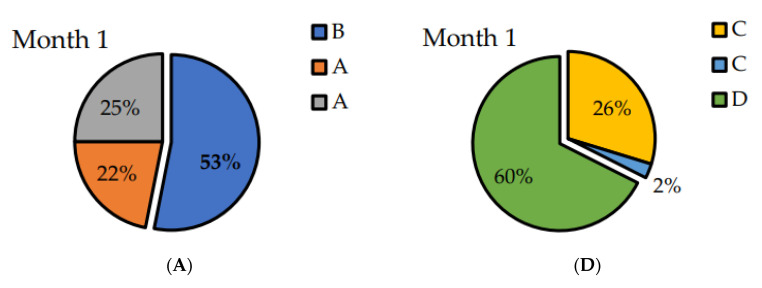
(**A**) Results of the first triangle test comparing P_Control (A) and P_ML samples (B) for month 1; (**B**) Results of the first triangle test comparing P_Control (A) and P_ML samples (B) for month 2; (**C**) Results of the first triangle test comparing P_Control (A) and P_ML samples (B) for month 3; (**D**) Results of the second triangle test comparing P_ML/LDPE_CAR (C) and P_ML samples (D) for month 1; (**E**) Results of the second triangle test comparing P_ML/LDPE_CAR (C) and P_ML samples (D) for month 2; (**F**) Results of the second triangle test comparing P_ML/LDPE_CAR (C) and P_ML samples (D) for month 3.

**Figure 6 polymers-14-03550-f006:**
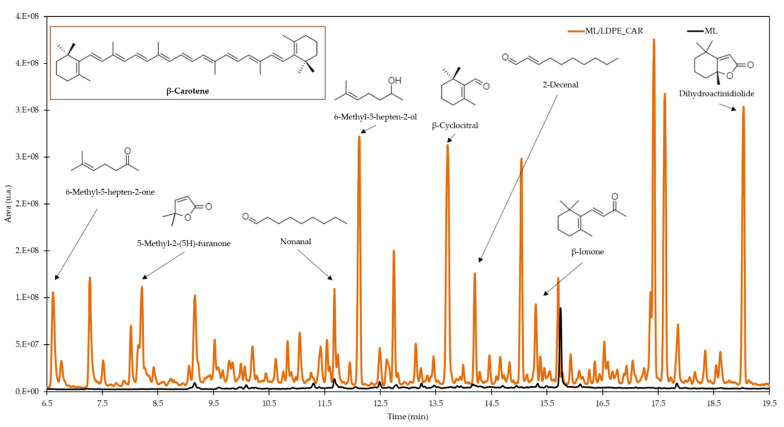
Chromatogram of volatile fraction of multilayer films. β-carotene derivatives in ML/LDPE_CAR are shown.

**Table 1 polymers-14-03550-t001:** Colour parameters (*L**, *a**, *b**, and Δ*E*) of monolayer and multilayer studied films ^#,†^.

	L*	a*	b*	ΔE
LDPE	99.0 ± 0.3 ^a^	0.03 ± 0.03 ^b^	0.1 ± 0.04 ^c^	0.27 ± 0.06 ^c^
ML	99.6 ± 0.2 ^a^	0.01 ± 0.01 ^b^	0.25 ± 0.05 ^c^	0.24 ± 0.09 ^c^
LDPE_CAR	75.2 ± 0.4 ^c^	15.2 ± 0.6 ^a^	77.4 ± 0.6 ^a^	82.1 ± 0.4 ^a^
ML/LDPE_CAR	98.15 ± 0.1 ^b^	−3.73 ± 0.09 ^c^	9.7 ± 0.4 ^b^	10.4 ± 0.4 ^b^

# Results expressed as the mean ± standard deviation of three replicates (α = 0.05). † Different superscripts in the same column indicate statistically significant different values (*p* < 0.05).

**Table 2 polymers-14-03550-t002:** Thermal properties of a laboratory scale films ^#,†^.

	T_ini_ (°C)	T_max_ (°C)	T_c_ (°C)	ΔH_c_ (J g^−1^)	T_m_ (°C)	ΔH_m_ (J g^−1^)	X_c_ (%)
LDPE	433 ± 2 ^a^	477 ± 1 ^b^	97± 0 ^a^	81.9 ± 1.7 ^a^	108 ± 0 ^a^	117.9± 1.6 ^b^	40 ± 1 ^a^
LDPE_CAR	341 ± 1 ^b^	482 ± 0 ^a^	97 ± 0 ^a^	78.5 ± 1.1 ^b^	108 ± 0 ^a^	120.5 ± 0.2 ^a^	37 ± 0 ^b^

# Results expressed as the mean ± standard deviation of three replicates (α = 0.05). † Different superscripts in the same column indicate statistically significant different values (*p* < 0.05).

**Table 3 polymers-14-03550-t003:** Thermal properties and oxygen induction time (OIt) of multilayers films ML and ML/LDPE_CAR ^#,†^.

	ML	ML/LDPE_CAR	Layer
T_ini_ (°C) *	401 ± 1 ^a^	397 ± 2 ^b^	LDPE
T_max_ (°C)	469 ± 2 ^b^	473 ± 1 ^a^	
T_c_ (°C) 1	93 ± 0 ^b^	95 ± 0 ^a^	
T_c_ (°C) 2	102 ± 0 ^a^	101 ± 0 ^b^	
ΔH_c_ (J g^−1^)	58.6 ± 0.8 ^b^	65.9 ± 0.8 ^a^	
T_m_ (°C) 1	102 ± 0 ^b^	105 ± 0 ^a^	
T_m_ (°C) 2	111 ± 0 ^a^	111 ± 0 ^a^	
ΔH_m_ (J g^−1^)	55.6 ± 1.9 ^b^	68.6 ± 1.7 ^a^	
X_c_ (%)	19 ± 1 ^a^	20 ± 1 ^a^	
T_m_ (°C) 1	182 ± 0 ^a^	176 ± 6 ^a^	EVOH
ΔH_m_ (J g^−1^)	6.27 ± 0.05 ^a^	3.4 ± 0.2 ^b^	
T_m_ (°C) 1	254 ± 0 ^a^	253 ± 0 ^b^	PET
ΔH_m_ (J g^−1^)	9.5 ± 0.4 ^a^	6.1 ± 1.1 ^b^	
OIt (min)	4.5 ± 0.3 ^b^	14.1 ± 0.3 ^a^	

^#^ Results expressed as the mean ± standard deviation of three replicates (α = 0.05). ^†^ Different superscripts in the same raw indicate statistically significant different values (*p* < 0.05).

**Table 4 polymers-14-03550-t004:** Overall migrations values (in mg dm^−2^) in ML and ML/LDPE_CAR films.

	EtOH 10% (*v*/*v*)	EtOH 95% (*v*/*v*) *	Isooctane *
ML	0.2 ± 0.1	0.09 ± 0.04	0.44 ± 0.02
ML/LDPE_CAR	1.6 ± 0.5	2.62 ± 0.16	3.32 ± 0.14

* Corrected.

## Data Availability

Not applicable.

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
