# Peer review of "Oxygen Scavenger and Antioxidant LDPE/EVOH/PET-Based Films Containing β-Carotene Intended for Fried Peanuts (Arachis hypogaea L.) Packaging: Pilot Scale Processing and Validation Studies"

_polymers, 2022, doi:10.3390/polym14173550_

Round 1

Reviewer 1 Report

In this paper, the authors developed an oxygen scavenger and antioxidant active packaging material for fried peanuts. The paper is interesting and can be published following minor revisions.

The authors used partially the template of the journal (the line numbers have been deleted).

All the acronyms have to be defined when used for the first time. See, for example, the name of the polymers, such as PE, PP, PET and so on.

There are some innovative techniques used to prepare active packaging, as the ones based on the use of supercritical fluids (doi: 10.1016/j.jcou.2019.06.012).

All the symbols used in equation (1) have to be defined.

Author Response

Alicante, 18th August 2022

Thank you for your positive remarks and for giving us the opportunity to revise and improve our manuscript polymers-1867426. We have carried out a careful revision by considering the referee’s recommendations and the reply to comments is in bold below in this letter. Page numbering in our answers refer to the revised manuscript.  

Please find enclosed the revised manuscript entitled: “Oxygen scavenger and antioxidant LDPE/EVOH/PET-based films containing β-carotene intended for fried peanuts (Arachis hypogaea L.). packaging: Pilot scale processing and validation studies.” which I am submitting as corresponding author for consideration of publication as a research paper in POLYMERS journal.

Reviewer 1

In this paper, the authors developed an oxygen scavenger and antioxidant active packaging material for fried peanuts. The paper is interesting and can be published following minor revisions. The authors used partially the template of the journal (the line numbers have been deleted).

Answer: The line numbers have been included.

All the acronyms have to be defined when used for the first time. See, for example, the name of the polymers, such as PE, PP, PET and so on.

Answer: Following the reviewer recommendations, all the acronyms have been defined when used for the first time.

There are some innovative techniques used to prepare active packaging, as the ones based on the use of supercritical fluids (doi: 10.1016/j.jcou.2019.06.012).

Answer: This reference has been included (línea 86).

All the symbols used in equation (1) have to be defined.

Answer: Symbols in equation (1) have been defined (lines 198-199).

Reviewer 2 Report

This study investigated the oxygen scavenger and antioxidant LDPE/PET-based films containing β-carotene intended for fried peanuts (Arachis hypogaea L.). packaging: Pilot scale processing and validation studies. The results are interesting and useful. However, the manuscript is not acceptable in its present form.

1. Rewrite the abstract. Authors are advised to add numerical data in this section.

2. The abbreviations should be defined first; Check the entire manuscript (abstract, introduction, etc.).

3. Section 2.2; How did the authors evaluate that 10% of beta-carotene is appropriate for this formulation?

4. Some references in the manuscript are shown as "Error! Reference source not found", please correct them.

5. At the end of the materials and methods section, the authors should add the "Statistical Analysis" section.

6. Table 1 is repeated.

7. For all tables, please use the letter "a" for the largest value.

8. Table 3; Thermal properties and oxygen induction time (OIt) of ...

9. Table 4, Figure 4, and Figure 5 did not mention in the text.

10. Section 3.3.2 (Hexanal content) should be 3.3.1. Also, report the range of hexanal content for ML and ML/LDPE_CAR films.

10. Section 3.2.2. (Sensorial Analysis) should be 3.3.2. Also, what did the author mean by "with different codifications" in line 3?

11. The last figure should be figure 6, not figure 1!

12. It is highly recommended that the authors report sensory analysis data.

Author Response

Alicante, 18th August 2022

Thank you for your positive remarks and for giving us the opportunity to revise and improve our manuscript polymers-1867426. We have carried out a careful revision by considering the referee’s recommendations and the reply to comments is in bold below in this letter. Page numbering in our answers refer to the revised manuscript.  

Please find enclosed the revised manuscript entitled: “Oxygen scavenger and antioxidant LDPE/EVOH/PET-based films containing β-carotene intended for fried peanuts (Arachis hypogaea L.). packaging: Pilot scale processing and validation studies.” which I am submitting as corresponding author for consideration of publication as a research paper in POLYMERS journal.

Reviewer 2

This study investigated the oxygen scavenger and antioxidant LDPE/PET-based films containing β-carotene intended for fried peanuts (Arachis hypogaea L.). packaging: Pilot scale processing and validation studies. The results are interesting and useful. However, the manuscript is not acceptable in its present form.

  1. Rewrite the abstract. Authors are advised to add numerical data in this section.

Answer: According the reviewer recommendations, abstract has been rewritten.

  1. The abbreviations should be defined first; Check the entire manuscript (abstract, introduction, etc.).
    Answer: Following the reviewer recommendations, all the acronyms have been defined when used for the first time.
  2. Section 2.2; How did the authors evaluate that 10% of beta-carotene is appropriate for this formulation?

Answer: When food is packaged under a modified atmosphere or in vacuum, a residual oxygen levels from 0.3 to 3.0 % remains inside the package. Then, as developed bags absorb 3.7 % of oxygen, β-carotene concentration at 10 wt. %. was appropriated for this application”. According to reviewer recommendation, the explanation has been completed at lines 469-472.

  1. Some references in the manuscript are shown as "Error! Reference source not found", please correct them.

Answer: Some errors in references have been corrected.

  1. At the end of the materials and methods section, the authors should add the "Statistical Analysis" section.

Answer: Statistical Analysis section has been added (lines 326-331).

  1. Table 1 is repeated.

Answer: Only one Table 1 is now present in the revised manuscript

  1. For all tables, please use the letter "a" for the largest value.

Answer: Following the reviewer recommendations, letter “a” has been used for the largest value in tables 1, 2, and 3.

  1. Table 3; Thermal properties and oxygen induction time (OIt) of ...

Answer: Following the reviewer recommendations, the table legend has been modified.

  1. Table 4, Figure 4, and Figure 5 did not mention in the text.

Answer: Following the reviewer recommendations, errors in figures and tables numbers have been corrected.

  1. Section 3.3.2 (Hexanal content) should be 3.3.1. Also, report the range of hexanal content for ML and ML/LDPE_CAR films.

Answer: Section number has been modified. Hexanal content was measured in peanuts packaged in bags produced from ML and ML/LDPE_CAR films (not directly in films!). The text was confusing. However, it has been corrected.

  1. Section 3.2.2. (Sensorial Analysis) should be 3.3.2. Also, what did the author mean by "with different codifications" in line 3?

Answer: Section number has been modified. The sentence “with different codifications” has been replaced by “coded randomly”.

  1. The last figure should be figure 6, not figure 1!

Answer: Yes, the last figure is figure 6.

  1. It is highly recommended that the authors report sensory analysis data.

Answer: Following the reviewer recommendations, sensory analysis data has been included in figure 5.

Reviewer 3 Report

The manuscript is focused on the production and characterization of an oxygen scavenger and antioxidant active packaging material for fried peanuts. The idea is fairly novel and the experimental design was generally well-done and well organized, and I recommend publication after the following corrections:

Title:

In material section, it was mentioned that a commercial film with a structure relation of PET/EVOH/PE was used in the present study. However, in the title (and elsewhere in the text) “LDPE/PET-based films” was used. Why is that?

Abstract:

The sentence “The obtained films were compared in terms of their colour, thermal properties, oxygen absorption capacity and overall migration tests confirming an antioxidant effect of β-carotene on the developed materials resulting in an increase in thermal stability and oxygen induction time (OIt)” is ambiguous. Please rewrite.

Replace the “Regarding the oxygen capacity” with “Regarding the oxygen absorption capacity”.

 “LDPE_CAR” and “ ML/LDPE_CAR” are not clear, and they must be defined earlier.

Introduction:

Some sentences in the introduction section like the followings had not any references. Please add the relevant references: “Other variables that affect lipid oxidation are temperature, water content, free fatty acid content, oxygen concentration, and the presence of prooxidant and antioxidant substances. In this line, tocopherols present in peanuts (mainly α- and γ-tocopherol homologues) act as an antioxidant barrier and protect unsaturated fatty acids from suffering oxidative damage. When peanuts are being oxidised, the concentration of some volatile compounds such as hexanal, a rancidity indicator, increase as it is a secondary metabolite of lipid oxidation. Consequently, hexanal analysis is a useful tool to monitor oxidative damage in peanuts

Correct “E. A. Abdel-Razik” (delete E. A.)

Butylated hydroxytoluene (BHT)” was defined twice. Delete the second description

Regarding the paragraphs started with “In the recent literature, some oxygen scavenger …” and “Consumer’s preferences have steadily evolved …”, it seems that a lot of unnecessary information was mentioned. Please revise.

Regarding “2,2’-azino-bis(3-ethylbenzothiazoline-6-sulfonic acid (ABTS)”, it seems that a “)” should be added.

The “2.10-4 g β-carotene/ mL” should be replaced by “2*10-4 g β-carotene/ mL

Why β-carotene was incorporated at 10 wt. % in this study?

Correct “2.3.1.β-. carotene Quantification

In the subheadings, start the second word with a lowercase letter; i.e. “Films Processing

In section 2.3.2. “the reference peanut” should probably be replaced by “the reference film”.

Section 2.3.3. the method of calculation of OIt must be described.

The term “ incorporated” in “were incorporated into 70 μL aluminium” should probably be replaced by “were added”.

Section 2.3.5. it seems that “)” must be added after (v/v)

Define “dm-2

Replace “ (QQQ)” with “(QMA)”

The phrase “Error! Reference source not found” must be corrected throughout the text.

2.4. the control was stored under what packing condition?

2.5.2. “by using” must be replaced by “using”

Add “, and” after verified

What are “global sensory differences “?!

Result and discussion:

Section 3.1. delete the extra table at the beginning of this section

How do you justify the b* and especially a* changes in ML/PE_CAR film compared to ML and LDPE_CAR films?

For evaluating the effect of β-carotene on the oxygen absorption capacity of the films, the addition of β-carotene to other plastic can be used.

Please discuss why ML film showed higher oxygen absorption capacity than single layer LDPE.

Table 3. correct “#,†” in the title.

Table 3. add a title for the fourth column (LDPE, EVOH, …)

Table 3. “the same column” must be replaced by “the same row”

Define “ABS”

Table 4. add the migration unit, and it seems that “*” was used in incorrect positions. Please revise the table.

What is the limit of acceptability of hexanal in the product? Please discuss in the text (section 3.3.2)

Regarding the triangular sensory tests, the method section is not clear (time of evaluation and the groups that were compared together, etc.). Please revise. Moreover, why the results of the second triangular sensory test were not presented like the first one?!

Author Response

Alicante, 18th August 2022

Thank you for your positive remarks and for giving us the opportunity to revise and improve our manuscript polymers-1867426. We have carried out a careful revision by considering the referee’s recommendations and the reply to comments is in bold below in this letter. Page numbering in our answers refer to the revised manuscript.  

Please find enclosed the revised manuscript entitled: “Oxygen scavenger and antioxidant LDPE/EVOH/PET-based films containing β-carotene intended for fried peanuts (Arachis hypogaea L.). packaging: Pilot scale processing and validation studies.” which I am submitting as corresponding author for consideration of publication as a research paper in POLYMERS journal.

Reviewer 3

The manuscript is focused on the production and characterization of an oxygen scavenger and antioxidant active packaging material for fried peanuts. The idea is fairly novel, and the experimental design was generally well-done and well organized, and I recommend publication after the following corrections:

Title: In material section, it was mentioned that a commercial film with a structure relation of PET/EVOH/PE was used in the present study. However, in the title (and elsewhere in the text) “LDPE/PET-based films” was used. Why is that?

Answer: According to reviewer recommendation, the title has been modified.

Abstract: The sentence “The obtained films were compared in terms of their colour, thermal properties, oxygen absorption capacity and overall migration tests confirming an antioxidant effect of β-carotene on the developed materials resulting in an increase in thermal stability and oxygen induction time (OIt)” is ambiguous. Please rewrite.

Replace the “Regarding the oxygen capacity” with “Regarding the oxygen absorption capacity”.

Answer: According the reviewer recommendations, abstract has been rewritten. The sentence has been modified accordingly (line 26).

 “LDPE_CAR” and “ ML/LDPE_CAR” are not clear, and they must be defined earlier.

Answer: These terms have been defined in the abstract section.

Introduction:

Some sentences in the introduction section like the followings had not any references. Please add the relevant references: “Other variables that affect lipid oxidation are temperature, water content, free fatty acid content, oxygen concentration, and the presence of prooxidant and antioxidant substances. In this line, tocopherols present in peanuts (mainly α- and γ-tocopherol homologues) act as an antioxidant barrier and protect unsaturated fatty acids from suffering oxidative damage. When peanuts are being oxidised, the concentration of some volatile compounds such as hexanal, a rancidity indicator, increase as it is a secondary metabolite of lipid oxidation. Consequently, hexanal analysis is a useful tool to monitor oxidative damage in peanuts”

Answer: According to reviewer recommendation, five references have been included in that paragraph.

Correct “E. A. Abdel-Razik” (delete E. A.)

Answer: The sentence has been modified accordingly (line 119).

“Butylated hydroxytoluene (BHT)” was defined twice. Delete the second description.

Answer: The second description has been deleted.

Regarding the paragraphs started with “In the recent literature, some oxygen scavenger …” and “Consumer’s preferences have steadily evolved …”, it seems that a lot of unnecessary information was mentioned. Please revise.

Answer: Both paragraphs have been revised accordingly.

Regarding “2,2’-azino-bis(3-ethylbenzothiazoline-6-sulfonic acid (ABTS)”, it seems that a “)” should be added.

Answer: The sentence has been modified accordingly (line 109).

The “2.10-4 g β-carotene/ mL” should be replaced by “2*10-4 g β-carotene/ mL”.

Answer: The sentence has been modified accordingly (line 116).

Why β-carotene was incorporated at 10 wt. % in this study?

Answer: When food is packaged under a modified atmosphere or in vacuum, a residual oxygen levels from 0.3 to 3.0 % remains inside the package. Then, as developed bags absorb 3.7 % of oxygen, β-carotene concentration at 10 wt. %. was appropriated for this application”. According to reviewer recommendation, the explanation has been completed at lines 469-472.

Correct “2.3.1.β-. carotene Quantification”.

Answer: The sentence has been modified accordingly (line 182).

In the subheadings, start the second word with a lowercase letter; i.e. “Films Processing”

Answer: All the subheadings have been revised.

In section 2.3.2. “the reference peanut” should probably be replaced by “the reference film”.

Answer: The sentence has been modified accordingly (line 196).

Section 2.3.3. the method of calculation of OIt must be described.

Answer: The method of calculation of OIt has been described (lines 217-219).

The term “ incorporated” in “were incorporated into 70 μL aluminium” should probably be replaced by “were added”.

Answer: The sentence has been modified accordingly (line 203 and line 214).

Section 2.3.5. it seems that “)” must be added after (v/v)

Answer: It has been revised accordingly.

Define “dm-2”

Answer: It refers to the migration of the compounds (mg) in a standard contact area of ​​1 dm2 and it has been clarified in the revised version of the manuscript (lines 251-253).

Replace “ (QQQ)” with “(QMA)”

Answer: It has been changed accordingly.

The phrase “Error! Reference source not found” must be corrected throughout the text.

Answer: Some errors in references have been corrected.

2.4. the control was stored under what packing condition?

Answer: According to reviewer recommendation, more information about packing condition of control samples has been added (lines 280-281).

2.5.2. “by using” must be replaced by “using”

Answer: It has been revised accordingly.

Add “, and” after verified

Answer: It has been revised accordingly (line 306).

What are “global sensory differences “?!

Answer: According to reviewer recommendation, “global sensory differences “ has been replaced with “overall perception” (line 310).

Result and discussion:

Section 3.1. delete the extra table at the beginning of this section

Answer: The extra table has been deleted.

How do you justify the b* and especially a* changes in ML/PE_CAR film compared to ML and LDPE_CAR films?

Answer: Changes in b* and especially a* values have been justified in lines 401-403.

For evaluating the effect of β-carotene on the oxygen absorption capacity of the films, the addition of β-carotene to other plastic can be used.

Answer: From the best of our knowledge, the oxygen absorption capacity of natural compounds such as β-carotene incorporated in a polymeric matrix had not been tested before (lines 467-467).

Please discuss why ML film showed higher oxygen absorption capacity than single layer LDPE.

Answer: Oxygen absorption capacity between LDPE_CAR and ML/LDPE_CAR film did not show significant differences (lines 461-462).

Table 3. correct “#,†” in the title.

Answer: It has been corrected accordingly,

Table 3. add a title for the fourth column (LDPE, EVOH, …)

Answer: The term layer has been added.

Table 3. “the same column” must be replaced by “the same row”

Answer: It has been corrected accordingly.

Define “ABS”

Answer: This term has been defined (line 120).

Table 4. add the migration unit, and it seems that “*” was used in incorrect positions. Please revise the table.

Answer: The migration units have been added.

What is the limit of acceptability of hexanal in the product? Please discuss in the text (section 3.3.2)

Answer: The limit of acceptability was no evaluated in this study. However, “considering P_ML presented an increment of 175% in hexanal content after 1 month of thermal treatment (Figure 4), we could conclude this increment is enough to produce considerable sensory changes.” (lines 543-545).

Regarding the triangular sensory tests, the method section is not clear (time of evaluation and the groups that were compared together, etc.). Please revise. Moreover, why the results of the second triangular sensory test were not presented like the first one?!

Answer: Following the reviewer recommendations, sensory test methodology has been extended (lines 308-314). Moreover, sensory analysis data has been included in figure 5.

Round 2

Reviewer 2 Report

The authors addressed all my suggested comments; this manuscript is now acceptable.

Reviewer 3 Report

The manuscript has been extensively revised and improved. So, it can be accepted in present form.